# Neuroprotective Effects of *Davallia mariesii* Roots and Its Active Constituents on Scopolamine-Induced Memory Impairment in In Vivo and In Vitro Studies

**DOI:** 10.3390/ph16111606

**Published:** 2023-11-14

**Authors:** Chung Hyeon Lee, Min Sung Ko, Ye Seul Kim, Ju Eon Ham, Jee Yeon Choi, Kwang Woo Hwang, So-Young Park

**Affiliations:** 1College of Pharmacy, Dankook University, Cheonan 31116, Republic of Korea; cndgus1995@naver.com (C.H.L.); dkrkekxn@naver.com (M.S.K.); yees_l@naver.com (Y.S.K.); 2College of Pharmacy, Chung-Ang University, Seoul 06974, Republic of Korea; hju980925@gmail.com (J.E.H.); choijeeyeon87@gmail.com (J.Y.C.)

**Keywords:** ethanol extract of *D. mariesii* roots, Aβ production, Aβ aggregation, scopolamine-induced memory impairment, flavonoids

## Abstract

Beta-amyloid (Aβ) proteins, major contributors to Alzheimer’s disease (AD), are overproduced and accumulate as oligomers and fibrils. These protein accumulations lead to significant changes in neuronal structure and function, ultimately resulting in the neuronal cell death observed in AD. Consequently, substances that can inhibit Aβ production and/or accumulation are of great interest for AD prevention and treatment. In the course of an ongoing search for natural products, the roots of *Davallia mariesii* T. Moore ex Baker were selected as a promising candidate with anti-amyloidogenic effects. The ethanol extract of *D. mariesii* roots, along with its active constituents, not only markedly reduced Aβ production by decreasing β-secretase expression in APP–CHO cells (Chinese hamster ovary cells which stably express amyloid precursor proteins), but also exhibited the ability to diminish Aβ aggregation while enhancing the disaggregation of Aβ aggregates, as determined through the Thioflavin T (Th T) assay. Furthermore, in an in vivo study, the extract of *D. mariesii* roots showed potential (a tendency) for mitigating scopolamine-induced memory impairment, as evidenced by results from the Morris water maze test and the passive avoidance test, which correlated with reduced Aβ deposition. Additionally, the levels of acetylcholine were significantly elevated, and acetylcholinesterase levels significantly decreased in the brains of mice (whole brains). The treatment with the extract of *D. mariesii* roots also led to upregulated brain-derived neurotrophic factor (BDNF) and phospho-cAMP response element-binding protein (p-CREB) in the hippocampal region. These findings suggest that the extract of *D. mariesii* roots, along with its active constituents, may offer neuroprotective effects against AD. Consequently, there is potential for the development of the extract of *D. mariesii* roots and its active constituents as effective therapeutic or preventative agents for AD.

## 1. Introduction

Alzheimer’s disease (AD), initially documented by the German physician Alois Alzheimer, is a neurodegenerative disease that predominantly affects individuals aged 65 and older [1]. It progressively impacts various cognitive functions, leading to impairments in speech, spatiotemporal orientation, language, and motor skills, primarily due to memory loss [2,3,4]. Presently, AD treatment is focused on mitigating clinical symptoms and delaying cognitive and memory decline; however, no specific treatments or drugs have been developed thus far [5,6,7]. The neuropathological characteristics of AD encompass neurofibrillary tangles, which are accumulations of hyperphosphorylated tau proteins found within neurons, and senile plaques, which result from the accumulation of beta-amyloid (Aβ) outside nerve cells [8,9,10]. Aβ, a major contributor to AD, induces structural and functional changes in neurons, culminating in the generation of oxidative stress, cytotoxicity, and inflammatory cytokines and chemokines, ultimately leading to neuronal cell death [11,12]. Aβ, comprising 38–43 amino acid peptides, is a metabolite produced by the cleavage of a protein derived from amyloid precursor protein (APP) by degrading enzymes, β-secretase, known as β-site APP-cleaving enzyme (BACE1), and γ-secretase, consisting of four or more protein complexes (presenilin 1 or 2, presenilin enhancer 2, anterior phaynx, or nicastrin) [13,14]. β-secretase cleaves the ectodomain, a domain of the membrane protein that extends into the extracellular space of APP, and divides it into an N-terminal extracellular domain called sAPPβ and a C-terminal cytoplasmic domain called CTFβ; sAPPβ is released from the cell, and the CTFβ is further cleaved by γ-secretase to produce 4 kDa Aβ [15,16,17,18,19,20]. The released Aβ monomers aggregate into oligomers and fibrils, leading to alterations in neuronal activity and the release of neurotoxic substances by microglial cells, eventually resulting in impaired synaptic function and brain damage [21]. Additionally, the accumulation of Aβ aggregates leads to the generation of free radicals, which causes the neuronal death observed in AD [22]. Therefore, reducing the production of neurotoxic Aβ and inhibiting its aggregation represents a promising therapeutic and preventive approach for AD [23].

Scopolamine is frequently employed to induce cognitive impairment in an AD-like experimental model, due to its ability to permeate the blood–brain barrier (BBB) [24]. When administered to male rats, scopolamine has been shown to increase the levels of Aβ proteins and APP mRNA expression [25]. Additionally, scopolamine promotes the hyperphosphorylation of tau protein by enhancing the activity of tau kinases, a key pathological hallmark of AD [26,27]. Moreover, scopolamine induces oxidative stress, mitochondrial dysfunction, and neuroinflammation [28,29,30,31]. Therefore, animals administered with scopolamine serve as a valuable experimental model for studying the alterations in AD pathogenesis [32].

Our previous study reported that the ethanol extract of *Dryopteris crassirhizoma* roots, a fern species in the family Dryopteridaceae, significantly reduced Aβ production in an in vitro assay [33], and its anti-amyloidogenic and memory-improving effects were confirmed in an in vivo scopolamine-induced model (unpublished data). As a result, we have conducted continuous screening of more than 50 ferns for their effectiveness in inhibiting Aβ production and/or aggregation. The roots of *Davallia mariesii* T. Moore ex Baker were identified as a promising candidate with anti-amyloidogenic effects. *D. mariesii* is a perennial fern belonging to the family Davalliaceae, distributed throughout Korea, Japan, Taiwan, and China [34]. It is distinct from other ferns in that the underground stem is exposed on the rocks, extending to the side, and bearing closely attached linear-lanceolate and shield-shaped scales [34]. *D. mariesii* is known for its diverse biological activities, including analgesic properties, the induction of osteogenesis, the promotion of osteoclast apoptosis, and the enhancement of osteoblast proliferation [35]. Additionally, *D. mariesii* is recognized for its antifungal, antioxidant, and anti-allergic effects [36,37,38,39]. Furthermore, *D. mariesii* has been reported to contain various chemical constituents such as lactones, flavanone, catechins, epiafzelechins, glucuronates, sesquiterpenes, and phenolic glycosides [40,41,42,43,44]. However, studies on Aβ production and/or aggregation have not been reported.

In this study, we tested the ethanol extract of *D. mariesii* (referred to as DME) to evaluate its effects on Aβ production and aggregation, which were determined by Western blot analysis and Thioflavin T (Th T) assay, respectively. Subsequently, DME underwent diverse column chromatography (CC) processes to isolate active constituents using an activity-guided isolation methodology. The structures of isolated compounds were elucidated based on NMR data. Additionally, we investigated the potential memory-improving effects of DME in an in vivo scopolamine-treated mouse model, through passive avoidance and Morris water maze tests. We also measured the levels of acetylcholine (ACh), acetylcholinesterase (AChE), brain-derived neurotrophic factor (BDNF), phospho-cAMP response element-binding protein (p-CREB), CREB, and Aβ deposits in the brains using an enzyme-linked immunosorbent assay (ELISA) and immunohistochemistry.

## 2. Results

### 2.1. DME Decreased the Levels of sAPPβ and β-Secretase in APP–CHO Cells

The potential cytotoxicity of DME was assessed using the MTT assay, which revealed that DME did not diminish the viability of APP-CHO cells up to 100 μg/mL (Appendix A). To quantify the amount of Aβ, the levels of sAPPβ were determined through Western blot analysis, because sAPPβ is generated by the cleavage of APP by β-secretase, and sAPPβ production is directly related to Aβ content [45]. Additionally, the level of β-secretase was also analyzed through Western blot analysis.

The butanol-partitioned fraction of *Dryopteris crassirhizoma* roots (50 μg/mL) extract served as a positive control (PC) [33]. As shown in Figure 1, DME significantly reduced sAPPβ production in a dose-dependent manner. Treatment with 100 μg/mL of DME decreased sAPPβ levels to only 20%, compared to the DMSO-treated control group. Moreover, the level of β-secretase also showed a significant dose-dependent reduction, in response to DME treatment. The application of 100 μg/mL of DME reduced β-secretase levels to a level close to that of the positive control, which was approximately 30% of the DMSO-treated control group.

### 2.2. The Solvent-Partitioned Fractions of DME Decreased the Levels of sAPPβ and β-Secretase

DME was fractionated based on the solvent polarity, resulting in four fractions: *n*-hexane (Hx), dichloromethane (DCM), ethyl acetate (EA), and water (DW). The potential cytotoxicity of these solvent-partitioned fractions was assessed using the MTT assay. The DCM fraction at 100 μg/mL demonstrated cytotoxicity, while Hx, EA, and DW fractions showed no cytotoxic effects at any concentrations tested (Appendix A). Therefore, a concentration of 50 μg/mL was chosen to investigate the effects of the fractions on sAPPβ production.

The effects of the solvent-partitioned fractions on the levels of sAPPβ and β-secretase were determined through Western blot analysis. The Hx, DCM, and EA fractions (50 μg/mL) significantly reduced sAPPβ production and the level of β-secretase (Figure 2). Although the effects of the EA and DCM fractions on sAPPβ and β-secretase levels were similar, the EA fraction was selected for further studies since it had a higher amount of EA fraction (29.1 g compared to 1.9 g for DCM), lower cytotoxicity, and ease of isolating active compounds.

### 2.3. Both DME and Its Solvent-Partitioned Fractions Decreased Aβ Aggregation and Enhanced the Disaggregation of Pre-Formed Aβ Aggregates

The potential effects of DME and its solvent-partitioned fractions on Aβ aggregation and the disaggregation of Aβ aggregates were assessed using the Th T assay. DME demonstrated a significant and dose-dependent reduction in Aβ aggregation. Among the four solvent-partitioned fractions, the DW fraction showed no significant effect, while the Hx, DCM, and EA fractions reduced Aβ aggregation in a dose-dependent manner when compared to the DMSO-treated Aβ-only control group (Figure 3A).

To evaluate the effects of DME and its solvent-partitioned fractions on the disaggregation of pre-aggregated Aβ, Aβ aggregates that had been formed over 24 h were incubated with DME and its solvent-partitioned fractions, at concentrations of 100, 20, and 4 μg/mL. The level of Aβ aggregation was subsequently measured using the Th T assay. DME and its solvent-partitioned fractions exhibited a positive, dose-dependent effect on Aβ disaggregation, compared to the DMSO-treated Aβ-only control group (Figure 3B). Specifically, incubation with 100 and 20 μg/mL of Hx and EA, as well as 100 μg/mL of DCM and DW, significantly enhanced the disaggregation of pre-formed Aβ aggregates.

### 2.4. Isolation of Active Compounds from DME and the Structure Elucidation

The EA fraction of DME underwent a series of fractionation steps using various open CC, with silica gel and Sephadex-LH 20 as stationary phases. This process followed the activity-guided isolation method outlined in the Materials and Methods. Consequently, four compounds were isolated with a purity exceeding 95%, and their structures (Figure 4) were determined by analyzing ^1^H- and ^13^C-NMR data and comparing them to relevant references [46,47,48,49,50]. These compounds were identified as (-)-epiafzelechin 3-*O*-β-D-allopyranoside (**1**), (2S)-5,7,3’,5’-tetrahydroxy-flavanone 7-*O*-β-D-glucopyranoside (**2**), (2S)-5,7,3’,5’-tetrahydroxyflavanone 7-*O*-neohesperidoside (**3**), and 7-*O*-methyl-epiafzelechin-(4α→8)-epiafzelechin 3-*O*-β-glucopyranoside (**4**).

### 2.5. Inhibitory Effects of the Isolated Compounds on sAPPβ and β-Secretase Levels

The potential cytotoxicity of compounds isolated from the ethyl acetate fraction of DME (10 and 50 μg/mL) was assessed using the MTT assay. None of these compounds reduced the viability of APP–CHO cells to less than 80% (Appendix A), and the subsequent studies were carried out using concentrations of 10 and 50 μg/mL.

Here, the effects of the compounds from DME on the levels of sAPPβ and β-secretase were determined through Western blot analysis. Compounds **1**–**4** reduced the production of sAPPβ (Figure 5). Notably, compound **2** at 50 μg/mL decreased sAPPβ levels to 35%, compared to the DMSO-treated control group, while compound 3 achieved a reduction to 40%. Additionally, all the compounds significantly lowered the levels of β-secretase, compared to the DMSO-treated control group. Compounds **1** and **2** at 50 μg/mL reduced β-secretase levels to just 20% of those in the DMSO-treated control group.

### 2.6. Compounds Isolated from DME Decreased Aβ Aggregation and Promoted the Disaggregation of Pre-Formed Aβ Aggregates

The effects of compounds isolated from DME on both Aβ aggregation and disaggregation of Aβ aggregates were evaluated using the Th T assay. As shown in Figure 6A, compounds **1**–**4** demonstrated significant reductions in Aβ aggregation compared to the DMSO-treated Aβ-only control group. Compound **2**, in particular, was the most effective among the four isolated compounds, decreasing Aβ aggregation to 45% of the level observed in the DMSO-treated Aβ-only control group.

To assess the impact of compounds isolated from DME on the disaggregation of pre-aggregated Aβ, Aβ pre-aggregated for 24 h in advance was incubated with the compounds (at concentrations of 50 and 10 μg/mL), and the extent of Aβ aggregation was measured using the Th T assay. As depicted in Figure 6B, all the compounds exhibited a positive effect on Aβ disaggregation, in comparison to the DMSO-treated Aβ-only control group. Compounds **1**, **3**, and **4** at 50 μg/mL showed modest effects in reducing Aβ aggregation by promoting the disaggregation of Aβ (to approximately 70% of the DMSO-treated Aβ-only control group). In contrast, compound **2** at 50 μg/mL significantly enhanced Aβ disaggregation, reducing Aβ aggregation to only 50% of the level observed in the DMSO-treated Aβ-only control group.

### 2.7. Evaluation of Changes in Body Weight and Behavioral Tests

Mice were randomly divided into five groups (n = 9): normal (N), control (P, saline + scopolamine), positive control (D, donepezil + scopolamine), DME (DME-L, DME-H, 200 or 500 mg/kg + scopolamine). After 1 week of acclimatization, the mice were administered with saline for N and P, donepezil for D, or 200 or 500 mg/kg DME for the DME group, for 28 days (Figure 7A). After the injection of scopolamine (intraperitoneal) on day 28 and 30, a passive avoidance test and a Morris water maze test were performed, respectively. The body weights of the animals were measured twice a week for 4 weeks (Figure 7B). The oral administration of DME did not affect the body weights compared to the control group, nor did it cause abnormal conditions in the mice.

The effect of DME on scopolamine-induced memory deficit was evaluated using the passive avoidance test on day 28. As shown in Figure 7C, scopolamine-treated mice (48.91 ± 4.9 s) showed a significant increase in retention time staying in the dark compartment, compared to the vehicle-treated normal mice (20.33 ± 6.1 s, *p* < 0.005). Conversely, the retention time of the 200 and 500 mg/kg DME-administered mice (38.17 ± 3.7 s and 35.05 ± 8.6 s, respectively) exhibited a tendency to decrease, but it was not statistically significant (*p* = 0.245 and 0.135, respectively). Treatment with donepezil (33.97 ± 8.1 s), a positive control, also reduced the retention time, but it was not statistically significant, either (*p* = 0.107).

To evaluate the effect of DME on the cognitive performance in scopolamine-induced memory impairment, the spatial memory of mice was assessed using the Morris water maze test. As shown in Figure 7D, the escape latency of the scopolamine-treated control group (65.59 ± 7.7 s) was significantly increased, compared to the vehicle-treated normal mice (29.05 ± 3.8 s, *p* < 0.005). Treatment of mice with 200 and 500 mg/kg DME (44.62 ± 9.1 s and 43.76 ± 8.1 s, respectively) decreased the escape latency, but it was not statistically significant (*p* = 0.829 and 0.714, respectively). The donepezil-treated group (38.47 ± 5.1 s) significantly lowered the escape latency (*p* < 0.05).

### 2.8. Inhibitory Effect of DME on the Levels of AChE and ACh in Mice

The effects of DME on ACh was evaluated by measuring the concentration of ACh in the whole brains of mice detected by ELISA (Figure 8A). The concentration of ACh was significantly decreased in the scopolamine-treated control group (146.15 ± 1.7 pg/mL) compared to the vehicle-treated normal group (217.11 ± 1.9 pg/mL) (*p* < 0.005). However, the administration of DME at 200 and 500 mg/kg restored the concentration of ACh (191.35 ± 3.9 pg/mL and 198.85 ± 3.1 pg/mL, respectively) and the concentrations of ACh in the DME-treated group were significantly higher than in the scopolamine-treated control group (*p* < 0.05). Donepezil also significantly increased the level of ACh (203.42 ± 3.2 pg/mL) compared to the scopolamine-treated control group (*p* < 0.005). The concentration of ACh in the donepezil-treated group was higher than in both of the DME-treated groups, but it was not statistically significant.

The activity of AChE was also determined by ELISA (Figure 8B). Treatment of scopolamine in the control group (138.49 ± 3.5 mU/mL) significantly increased the AChE activity, compared to the vehicle-treated normal group (103.21 ± 3.3 mU/mL) (*p* < 0.05). Conversely, the administration of DME at 200 and 500 mg/kg (90.61 ± 2.7 mU/mL and 84.71 ± 2.8 mU/mL, respectively) decreased the activity of AChE, and it was significantly lower than the scopolamine-treated control group (*p* < 0.005). The donepezil treatment (106.89 ± 1.8 mU/mL) also decreased the AChE activity (*p* < 0.05).

### 2.9. Effect of DME on Aβ Deposition, BDNF, and CREB Phophorylation

Aβ deposition, in the form of senile plaques, is a pathological hallmark of AD. Immunohistochemical examinations of experimental animal brain sections in the hippocampal regions were performed to detect Aβ deposition (Figure 9A). The control group treated with scopolamine exhibited an obvious accumulation of Aβ plaques in their brains (indicated by arrows), but the vehicle-treated normal group did not show any accumulation. Importantly, treatment with 200 and 500 mg/kg of DME reduced the deposition of Aβ, compared to the scopolamine-treated control group.

The expression of BDNF was dramatically reduced in the scopolamine-treated control group, compared to the vehicle-treated normal group (*p* < 0.005). However, the administration of DME at 500 mg/kg restored BDNF expression compared to the scopolamine-treated control group (*p* < 0.005). The administration of 200 mg/kg DME showed a tendency to increase BDNF expression, but it was not statistically significant. Additionally, the phosphorylation of CREB (p-CREB, p-Ser133) decreased after exposure to scopolamine in the scopolamine-treated control group (*p* < 0.005). However, mice treated with 500 mg/kg DME exhibited increased p-CREB expressions, compared to the scopolamine-treated control group (*p* < 0.005, Figure 9B), while DME at 200 mg/kg had minor effects on the expression of p-CREB. The expression of CREB in all groups was not significantly different from one another. The increases in BDNF and p-CREB were also observed in the donepezil-treated positive control group (*p* < 0.05).

## 3. Discussion

Aβ is produced through the cleavage of APP by two proteases, β- and γ-secretase [51]. The aggregation of Aβ monomers leads to the formation of Aβ oligomers and fibrils, which, in turn, promotes the generation of free radicals that react with proteins and lipids. This reaction compromises membrane integrity and damages the sensitivity of crucial enzymes necessary for neuronal function [12]. Continuous Aβ aggregation also triggers a chronic response from the innate immune system, activating microglia and resulting in neuroinflammation [13]. This, in turn, amplifies microglia-mediated neuronal death and the loss of neuronal synapses, which significantly contributes to the pathogenesis of AD [52]. In the present study, DME significantly reduced Aβ production and aggregation in an in vitro study. Additionally, DME exhibited a tendency to improve cognitive function in a scopolamine-treated mouse model, accompanied by a reduced ACh and AChE, as well as an increased expression of BDNF and p-CREB in the brain. These results demonstrate that DME may possess neuroprotective effects against AD.

Moreover, this study represents the first report to the inhibitory effects of DME and its active constituents, namely (-)-epiafzelechin-3-*O*-β-D-allopyranoside (**1**), (2S)-5,7,3’,5’-tetrahydroxy-flavanone 7-*O*-β-D-glucopyranoside (**2**), (2S)-5,7,3’,5’-Tetrahydroxy-flavanone 7-*O*-hesperidoside (**3**), and 7-*O*-methyl-epiafzelechin-(4α→8)-epiafzelechin-3-*O*-β-glucopyranoside (**4**), on Aβ production and aggregation. The treatment with DME and its isolated compounds significantly reduced Aβ production, accompanied by a decrease in the levels of β-secretase, which is an enzyme crucial for Aβ production from APP (Figure 1 and Figure 5). This result suggests that the reduced Aβ production by DME and its compounds could be attributed to the suppression of β-secretase. Furthermore, DME and its isolated compounds also significantly reduced Aβ aggregation. Multiple studies have proposed that the presence of an o-quinone moiety in flavonoids or catechol-containing phenolic compounds mediated the covalent adduct formation with amyloid proteins [53,54]. The compounds from DME, with the electrophilic carbonyls within o-quinone intermediates, could form covalent adducts with the nucleophilic side chain thiols and amines of Aβ [55]. Covalent adduct formation between the compounds and amyloid proteins may have a significant impact on reducing Aβ aggregation. In addition, the effects of DME and its active compounds on Aβ aggregation could be confirmed using TEM imaging.

Some flavonoids, such as (-)-epigallocatechin-3-gallate [56] and quercetin [57], are known to have beneficial effects on AD. Flavonoids are secondary metabolites known for their anti-cancer, antioxidant, anti-inflammatory, anti-microbial, and immunomodulatory properties. Additionally, flavonoids have been reported to have the potential to mitigate cognitive decline, restore memory function, and delay the onset of conditions associated with AD [58,59]. The neuroprotective effects of flavonoids are attributed to their antioxidant properties. In fact, oxidative stress reduces the activity of α-secretase and promotes the activation of redox-sensitive cell signaling pathways, including JNK, which in turn enhances the expression of β- and γ-secretases [60,61]. Thus, flavonoids may counteract the detrimental effects of oxidative stress in the progression of AD. Furthermore, flavonoids are known to exhibit inhibitory effects on AChE and BChE, with their inhibitory potency being influenced by the number and position of hydroxy groups in the phenyl ring. These hydroxy groups can form multiple hydrogen bonds in the enzyme’s active site [62]. Moreover, phenolic hydroxy groups in flavonoids form hydrogen bonds with both carboxyl and amine groups in the Aβ peptide, thereby disrupting the β-sheet structures of Aβ fibrils and inhibiting Aβ aggregation [63]. Therefore, DME, with flavonoids as active constituents, may be beneficial for the treatment or prevention of AD through multiple mechanisms of action.

The neuroprotection provided by flavonoids has the capacity to penetrate the central nervous system by crossing the BBB [64]. Certain flavonoids, such as naringenin, quercetin, anthocyanins, epicatechins, and their glucuronidated conjugates have been observed to traverse the BBB in in vitro models, using brain endothelial cell lines and ECV304 monolayers co-cultured with C6 glioma cells [64]. However, the permeability of some flavonoids in vivo can be influenced by factors like their lipophilicity and interactions with efflux transporters [65]. Therefore, it is essential to study the permeability of the isolated flavonoids from DME.

The effects of DME on memory Improvement were assessed in mice with scopolamine-induced memory impairment using both the passive avoidance test, commonly employed to measure long-term memory [66], and the Morris water maze test, primarily utilized to evaluate hippocampus-dependent spatial learning ability [67,68,69]. Scopolamine, a muscarinic receptor antagonist, is a frequently used treatment because it induces memory impairment [70] due to cholinergic deficits, which are correlated with the severity of AD [71]. For instance, moscatilin derived from *Dendrobrium loddigesii* improved cognition and behavior in scopolamine-induced mice by downregulating p-Tau expression, one of the neuropathological hallmarks of AD [72]. Additionally, triterpenoids from *Cimicifuga dahurica* enhanced the cognitive performance of mice with scopolamine-induced deficits in behavioral tests and the expression of memory-related proteins [73]. In this study, the administration of DME for 28 days exhibited a tendency to ameliorate scopolamine-induced learning impairment in both the passive avoidance test and Morris water maze test, although this improvement did not reach statistical significance. However, the observed trend towards an enhanced behavioral performance may be attributed to improved cholinergic function (inhibited AChE activity and increased ACh levels) resulting from DME treatment.

Acetylcholine, a neurotransmitter secreted by cholinergic neurons, and AChE, the enzyme that degrades ACh, are important for memory and learning ability [74,75]. Increased AChE activity and decreased ACh content are known to be associated with cognitive decline in AD patients [76]. In this study, the concentration of ACh in the scopolamine-treated group was significantly decreased, and the activity of AChE was significantly increased, compared to the whole brains of the normal group. Nevertheless, the treatment of DME reversed the effects of scopolamine on the levels of ACh and AChE. This result indicates that DME may suppress AChE activity and increase ACh levels in the scopolamine-treated cognitive impaired mice, and attributes, at least in part, to the tendency of cognitive improvement with the treatment of DME.

BDNF promotes the survival and differentiation of developing neurons, induces synaptic transmission, and regulates synaptic plasticity in the adult brain [77]. A reduced expression of BDNF is associated with progressive neuron atrophy in AD patients [78]. In cortical neurons, Ca^2+^ influx promotes the phosphorylation of CREB, which, in turn, activates BDNF transcription [79]. Therefore, the phosphorylation of CREB and the upregulation of BDNF expression could be beneficial for learning and memory in AD [80,81]. For instance, *Ishige foliacea* extract enhanced memory function in a mouse model with scopolamine-induced memory deficits, by activating BDNF-tropomyosin receptor kinase B (TrkB)-phosphorylated extracellular signal-regulated kinase (ERK) signaling in the hippocampus [82]. In our study, scopolamine significantly reduced the phosphorylation of CREB and the levels of BDNF, but the administration of DME restored the levels of BDNF and p-CREB. These results indicate that DME has beneficial effects on cognitive function, partially through the upregulation of BDNF and p-CREB.

To confirm the neuroprotective effects of DME and its active constituents, it is essential to validate their benefits through an in vivo animal study involving AD-transgenic mice. Moreover, the permeability of DME across the BBB should be investigated, along with its effects on the expression of APP.

## 4. Materials and Methods

### 4.1. Preparation of D. mariesii Extract (DME)

The roots of *D. mariesii* were purchased from the Jeju five-day traditional market in July of 2020. A voucher specimen (120000059889) was deposited in the Pharmacognosy Laboratory of College of Pharmacy at Dankook University (Cheonan, Republic of Korea). The dried and pulverized roots of *D. mariesii* (3 kg) were extracted with 80% ethanol (20 L, three times) at room temperature. The filtrate was concentrated under vacuum to obtain 235.5 g of DME, which was subsequently utilized for further studies.

### 4.2. Isolation of Active Components

DME was partitioned based on solvent polarity, and four layers, including Hx (26.3 g), DCM (1.6 g), EA (29.1 g), and DW (50.0 g) layers, were obtained. The EA layer, which showed the highest activity in reducing the levels of sAPPβ and β-secretase, was subjected to open CC using silica gel (70–230 mesh, Watchers, Tokyo, Japan) and Sephadex LH-20 (GE Healthcare, Danderyd, Sweden) as stationary phases. A total of four compounds were isolated, and their structures were elucidated based on NMR data (Center for Bio-Medical Engineering Core Facility at Dankook University, Cheonan, Republic of Korea).

### 4.3. Cell Culture and MTT Assay

APP–CHO cells, namely Chinese hamster ovary cells stably expressing amyloid precursor protein (APP) [33], were grown in a RPMI 1640 (Welgene, Gyeongsan-si, Republic of Korea) medium with 10% heat-inactivated FBS (Gibco, Grand Island, NY, USA) in the presence of geneticin (Thermo Fisher Scientific, Waltham, MA, USA) (50 μg/mL). The cells were incubated in a humidified atmosphere with 5% CO_2_ at 37 °C. Cell viability was measured by the MTT (Biosesang, Yongin-si, Republic of Korea) assay. In brief, APP–CHO cells were plated in a 96-well plate with a density of 2 × 10^4^ cells in 100 μL of medium per well. The plate was incubated at 37 °C overnight to allow the cells to adhere to the plate. Before the cells were treated with test samples, the medium was replaced with RPMI 1640 without FBS for 1 h. Test samples diluted in DMSO (Biosesang) were added to individual wells, and the plates were incubated at 37 °C for 24 h. After the incubation of cells with 10 μL of 5 mg/mL MTT solution at 37 °C for 3 h, the medium was removed, and 100 μL of DMSO was added to each well. Plates were incubated at room temperature for 30 min to dissolve the MTT formazan crystals, and then the absorbance was measured at 540 nm using an E-max precision microplate reader (Epoch, BioTek, Winooski, VT, USA). Each experiment was performed in triplicate.

### 4.4. Western Blot Analysis

APP–CHO cells were plated in a six-well plate at a density of 6 × 10^5^ cells in 1000 μL of medium per well. The plates were incubated at 37 °C overnight to allow the cells to adhere to the plates. Subsequently, the medium was replaced with RPMI 1640 without FBS for 1 h, and the cells were then treated with test samples at 37 °C. After 24 h, the cells were washed with PBS and lysed in Laemmli sample buffer. The collected cell lysates were boiled at 100 °C for 10 min. For the Western blot analysis, 20 μg of proteins were separated, using 7.5% denaturing polyacrylamide gel electrophoresis (SDS-PAGE) based on protein size. They were then transferred to a PVDF membrane using Trans-Blot^®^ TurboTM (Biorad, Hercules, CA, USA). The membrane was blocked with a blocking buffer (5% skim milk in PBS) for 1 h and subsequently incubated with primary antibodies overnight at 4 °C. The primary antibodies used included sAPPβ antibody (1:1000, Immuno-Biological Laboratories, Fujioka, Japan), BACE1 antibody (1:1000, EMD Millipore, Burlington, MA, USA), and α-tubulin (25,000:1, Sigma Aldrich, Saint Louis, MO, USA). After washing the membrane with PBS-T (0.1% Tween 20 in PBS) three times for 10 min each, secondary antibodies conjugated to horseradish peroxidase (1:2500, Bio-Rad) were added for 1 h. Protein detection was performed using ECL solution reagent (Advansta, Menlo Park, CA, USA). The visualization of proteins was carried out with the ChemiDocTM XRS+ (Bio-Rad). For the quantification, the signal intensity of the band was measured using Bio-Rad software (version 5.2.1) and normalized by the signal intensity of the α-tubulin band. The results were expressed as a percentage of the DMSO-treated control group.

### 4.5. Thioflavin T (Th T) Assay

To evaluate the aggregation of Aβ to oligomers and fibrils, the Th T assay was performed. The Aβ1-42 (GL Biochem, Shanghai, China) was dissolved in DMSO at 1 mg/mL, and test samples were also dissolved in DMSO. To monitor the effects of test samples on Aβ aggregation, 20 μM of Aβ1-42 was incubated with various concentrations of test samples at 37 °C for 24 h. Then, 3 μM of Th T (Sigma Aldrich) was added, and fluorescence was measured after 30 min using an E-max precision microplate reader (BioTek) with excitation at 442 nm and emission at 485 nm. The Aβ treated with DMSO only was used as a control, and each assay was performed in triplicate. To monitor the disaggregation effects of test samples on the pre-aggregated Aβ, Th T assays were performed. Briefly, 20 μM of Aβ1-42 was incubated at 37 °C for 24 h separately. After that, various concentrations of test samples were added and incubated at 37 °C for additional 24 h. Then, 3 μM of Th T was added, and fluorescence was measured after 30 min using an E-max precision microplate reader (BioTek), with excitation at 442 nm and emission at 485 nm. The Aβ treated with DMSO only was used as a control and each assay was performed in triplicate.

### 4.6. Animals and Drug Administration

ICR mice (male, 6 weeks old) were purchased from Hana-Biotech (Seongnam-si, Korea), maintained on a 12 h light/12 h dark (lights on at 7 AM) cycle at a temperature of 23 ± 2 °C with the relative humidity at 60%, with regular daily diets and clean water ad libitum. Following a one-week acclimatization period, the mice were divided into five groups (n = 9/group); vehicle (N), scopolamine only (5 mg/kg, C), donepezil (0.75 mg/kg bw/day + scopolamine, P), DME 200 mg/kg bw/day + scopolamine (DME-L), and DME 500 mg/kg bw/day + scopolamine (DME-H). The mice were orally administered with either saline (N, C), donepezil (P) or DME for 28 days. To induce memory impairment, 5 mg/kg of scopolamine dissolved in sterilized saline was intraperitoneally injected 30 min after the final administration of saline, donepezil, or DME on day 28 and 29, and the passive avoidance test and Morris Water maze test were conducted (Figure 7A). Experimental procedures involving the mice were approved on 16 September 2021, under the reference number SEMI-21-012 by the Institutional Animal Care and Use Committee of SouthEast Medi-chem Institute.

### 4.7. Passive Avoidance Test

The passive avoidance test was carried out using two equally sized compartments (25 cm × 15 cm) with an electrifiable grid floor, and two compartments were separated by a guillotine door. The mice were initially placed in the light compartment, but they entered the dark compartment in accordance with the inherent tendency of rodents. When the mice moved into the dark compartment, the door closed automatically, and an electric foot shock (0.5 mA) for a 3 s duration was delivered through the grid floor. The experiment was conducted 24 h after the training period. For the passive avoidance test, the mice were once again placed in the dark compartment, and the retention time, which was the duration they remained in the dark compartment after the door was opened, was measured within 60 s.

### 4.8. Morris Water Maze Test

The Morris water maze experiment was conducted in a circular pool (diameter 100 cm, height 30 cm) with water maintained at the temperature 20 ± 2 °C. Black ink was added into the water to make the platform invisible. The water maze test area was divided into four quadrants, and the escape platform (10 cm diameter and 26 cm height) was placed 1 cm below the surface of the water in the center of one quadrant. The location of the intake point varied by section every day. The position of the platform was fixed in one place for the duration of the experiment. On the first day of the experiment, the training phase was initiated. If a mouse reached the platform, it was allowed to remain on the platform for 10 s. If the mouse did not find the platform within 120 s, the experiment was halted. In this case, the mouse was placed on the platform for 10 s to help it remember, and then the next experiment was carried out. On the second day of the experiment, the time taken for the mouse to find and climb the platform (escape latency) was measured. All experiments had a maximum duration of 120 s.

### 4.9. Determination of Acetylcholine (ACh) Level and Acetylcholinesterase (AChE) Activity

The mice were decapitated under ether anesthesia after 12 h of starvation, and their brains were removed and weighed. Subsequently, the brain was rinsed with ice-cold PBS and homogenized in 500 μL of PBS using a tissue homogenizer on ice. After two freeze–thaw cycles, the homogenates were centrifuged at 1500× *g* at 4 °C for 15 min. The supernatants were either used to measure the levels of ACh or AChE or stored at −80 °C for further use. The whole brains were utilized as tissue samples for measuring the levels of ACh and AChE. The Mouse ACh ELISA Kit (Cat. no #MBS160373, Mybiosource, San Diego, CA, USA) and Mouse AChE ELISA Kit (Cat, no #MBS721845, Mybiosource) were used to determine ACh level and AChE activity, respectively. To measure the ACh level, 40 μL of working solution was dispensed in a 96-well plate, and then 10 μL of anti-ACh antibody was added. In addition, 50 μL of the streptavidin–HRP reagent was dispensed, mixed well, sealed, and incubated at 37 °C. After washing the plate, 50 μL each of substrate solutions A and B were added, shielded from light, and incubated at 37 °C for about 10 min. After dispensing 50 μL of the stop solution, the absorbance was measured at 450 nm (BioTek). To measure the AChE activity, 100 μL of working solution was dispensed in a 96-well plate, and then 10 μL of balance solution was added. Then, 50 μL of the conjugate reagent was added, mixed well, sealed, and incubated at 37 °C. After washing the plate, 50 μL each of substrate A and B reagent were added, shielded from light, and incubated at 37 °C for 20 min. After dispensing 50 μL of the stop solution, the absorbance was measured at 450 nm (BioTek).

### 4.10. Immunohistochemistry (IHC) Staining Analysis

The brain tissue excised during autopsy was fixed in 10% formalin for 24 h. Subsequently, it was sequentially transferred to 10% and 20% sucrose solutions for 12 h each, and then stored in a 30% sucrose solution for 3 days. Serial paraffin sections (30 μm) were prepared, followed by deparaffinization and rehydration using a descending ethanol series (100%, 80%, 50%, H_2_O). The sections were incubated in 0.3% H_2_O_2_ in PBS for 30 min to quench endogenous peroxidase activity. They were immersed in a blocking solution for 1 h and incubated with the primary antibodies containing 1% bovine serum albumin for 1 h at room temperature. The primary antibodies used included anti-p-CREB (1:500, Novus, Littleton, CO, USA), anti-CREB (1:500, Novus), anti-BDNF (1:500, Novus), and anti-β-amyloid antibody (1:500, Novus). To visualize immunoreactivity, the sections were treated with avidin–biotin complex reagents (Vectastain ABC kit, Vector Laboratories, Newark, CA, USA) for 1 h at room temperature, followed by incubation with 3,3-diaminobenzidine tetrahydrochloride (DAB) and 0.01% H_2_O_2_ for 3 min. After rinsing, the sections were dehydrated using an ethanol series, followed by xylene, and then mounted. Histological images were observed and photographed at 100 X magnification using a standard optical microscope (Nikon, E600, Tokyo, Japan), and immune-positive cells were quantified with Image J Software (1.53k version) (National Institutes of Health, Bethesda, MD, USA)

### 4.11. Statistical Analysis

Data are presented as mean ± SD for the in vitro study and mean ± SE for the in vivo animal study. For the statistical analysis, a variance test using Levene’s test was conducted. If the resulting p-value exceeded 0.05, we accepted the assumption of homoscedasticity. Subsequently, two or more group comparisons were assessed by a one-way analysis of variance, followed by the Fisher’s least significant difference test (SPSS version 27.0, Armonk, NY, USA). However, if the p-value from the Levene’s test was less than 0.05, we rejected the assumption of homoscedasticity, indicating the presence of heteroscedasticity. In such cases, two or more group comparisons were evaluated using the Kruskal–Wallis test and the Mann–Whitney U-test (Jamovi software version 2.3.28). Statistically significant differences between values were considered to be present when the *p* value was below 0.05 (* *p* < 0.05).

## 5. Conclusions

In this study, DME and its active constituents, which include (-)-epiafzelechin-3-*O*-β-D-allopyranoside (**1**), (2S)-5,7,3’,5’-tetrahydroxy-flavanone 7-*O*-β-D-glucopyranoside (**2**), (2S)-5,7,3’,5’-tetrahydroxy-flavanone 7-*O*-hesperidoside (**3**), and 7-*O*-methyl-epiafzelechin-(4α→8)-epiafzelechin-3-*O*-β-glucopyranoside (**4**), significantly reduced Aβ production, the levels of β-secretase, and Aβ aggregation in in vitro assays. Moreover, the administration of DME demonstrated a tendency to attenuate memory impairment in an in vivo AD-like model induced by scopolamine, which was accompanied by a reduction in Aβ deposits in the brains. Additionally, the potential memory-improving effects of DME were associated with an increase in memory-related proteins (BDNF and p-CREB) and an enhancement in cholinergic function (ACh and AChE). This is the first report suggesting that the extract of *D. mariesii* roots may be beneficial in alleviating the symptoms of AD. Therefore, these results indicate that DME and its active constituents have the potential to be developed as natural supplements or new drugs for AD therapeutics and prevention.

## Figures and Tables

**Figure 1 pharmaceuticals-16-01606-f001:**
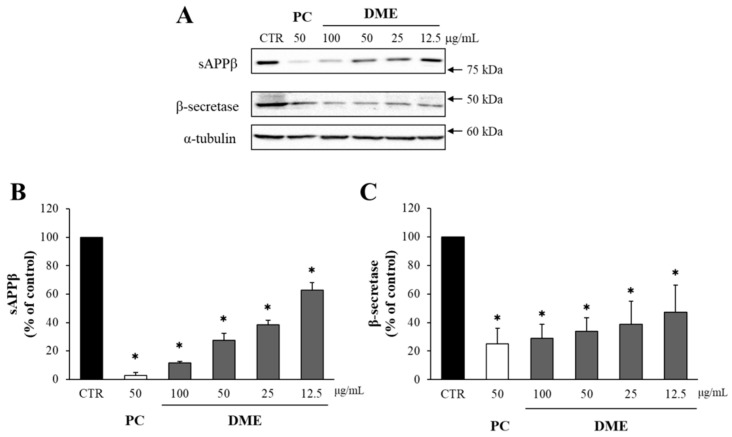
The effect of DME on sAPPβ and β-secretase. (**A**) The levels of sAPPβ and β-secretase in APP–CHO cells treated with various concentrations (100, 50, 25, and 12.5 μg/mL) of DME were determined by Western blot analysis. The graphs show the levels of sAPPβ (**B**) and β-secretase (**C**), compared to the DMSO-treated control group. Values are expressed as a percentage of DMSO-treated control group. All data represent the means ± SD of three different experiments. * *p* < 0.05, significantly different from DMSO-treated control group (CTR: DMSO-treated control, DME: ethanol extract of *D. mariesii* roots, PC: positive control).

**Figure 2 pharmaceuticals-16-01606-f002:**
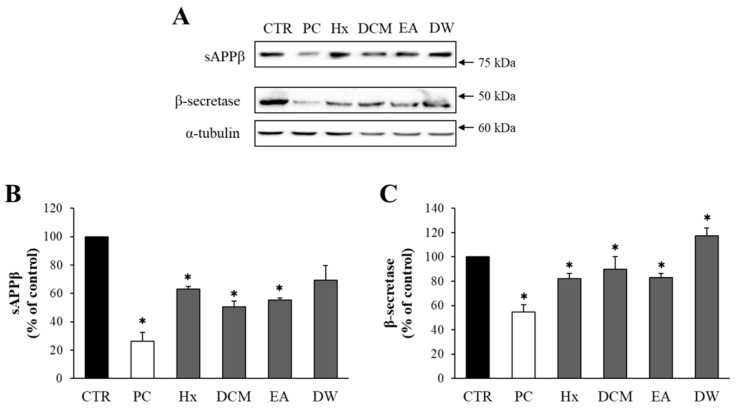
Effects of the solvent-partitioned fractions on the production of sAPPβ and the level of β-secretase. (**A**) The levels of sAPPβ and β-secretase from APP–CHO cells treated with 50 μg/mL of the solvent-partitioned fractions were determined by Western blot analysis. (**B**,**C**) The graphs show the levels of sAPPβ and β-secretase. Values are expressed as a percentage of DMSO-treated control group. All data represent the means ± SD of three different experiments. * *p* < 0.05, significantly different from DMSO-treated control group (Hx: *n*-hexane, DCM: dichloromethane, EA: ethyl acetate, DW: water).

**Figure 3 pharmaceuticals-16-01606-f003:**
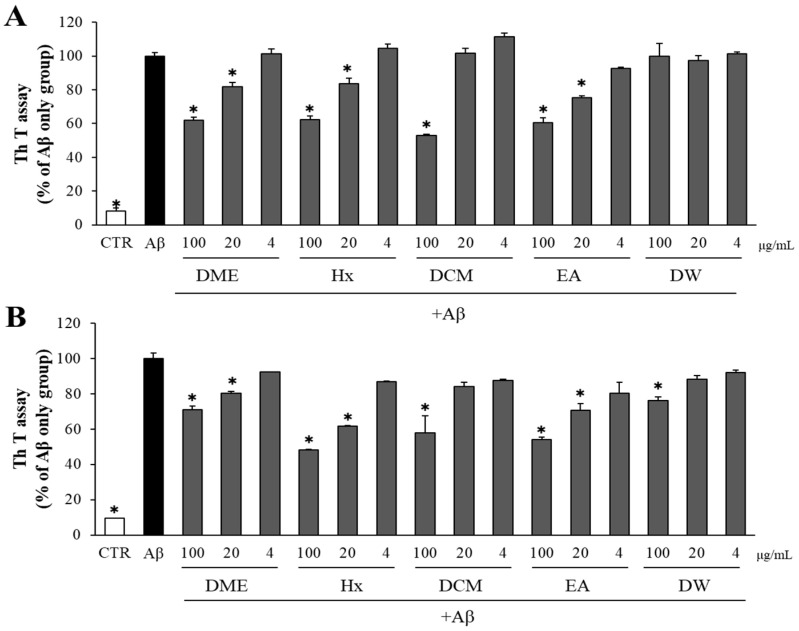
Inhibitory effect of DME and solvent-partitioned fractions on Aβ aggregation/disaggregation. (**A**) Aβ was incubated with 100, 20, and 4 μg/mL of DME and solvent-partitioned fractions (Hx: *n*-hexane, DCM: dichloromethane, EA: ethyl acetate, DW: water). After 24 h, the Aβ aggregation was determined by Th T assay. (**B**) Aβ pre-aggregated for 24 h was incubated with DME and solvent-partitioned fractions (100, 20, and 4 μg/mL). After additional 24 h, the Aβ aggregation was determined by Th T assay. All data represent the means ± SD of three different experiments, * *p* < 0.05, significantly different from Aβ-only group.

**Figure 4 pharmaceuticals-16-01606-f004:**
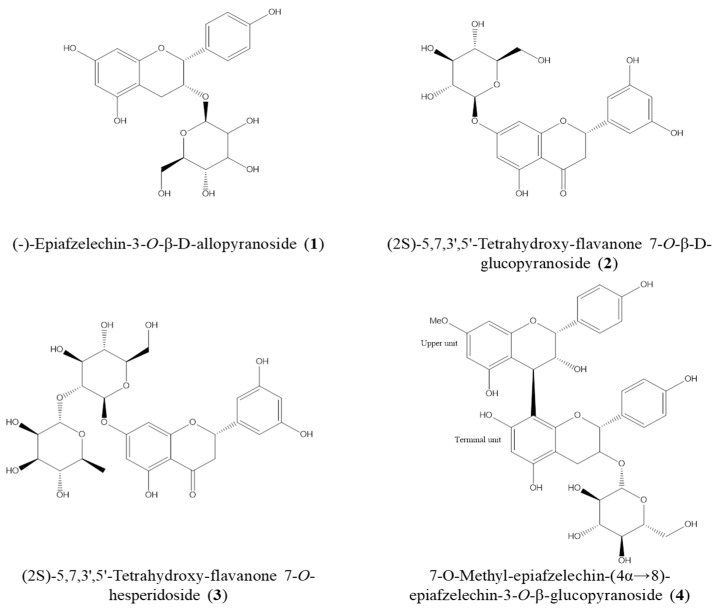
The chemical structures of the isolated compounds from the ethyl acetate fraction of DME.

**Figure 5 pharmaceuticals-16-01606-f005:**
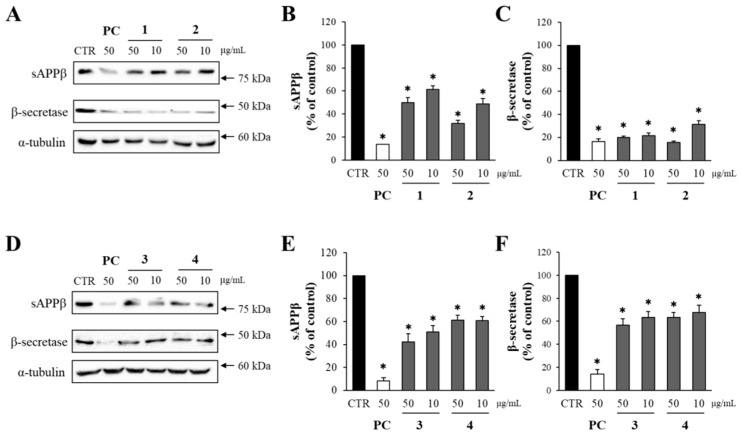
The effects of compounds **1**–**4** on the production of sAPPβ and the level of β-secretase. (**A**,**D**) The levels of sAPPβ and β-secretase from APP–CHO cells treated with compounds **1**–**4** (50 and 10 μg/mL) were determined by Western blot analysis. (**B**,**C**,**E**,**F**) The graphs show the levels of sAPPβ and β-secretase. Values are expressed as a percentage of DMSO-treated control group. All data represent the means ± SD of three different experiments. * *p* < 0.05, significantly different from DMSO-treated control group.

**Figure 6 pharmaceuticals-16-01606-f006:**
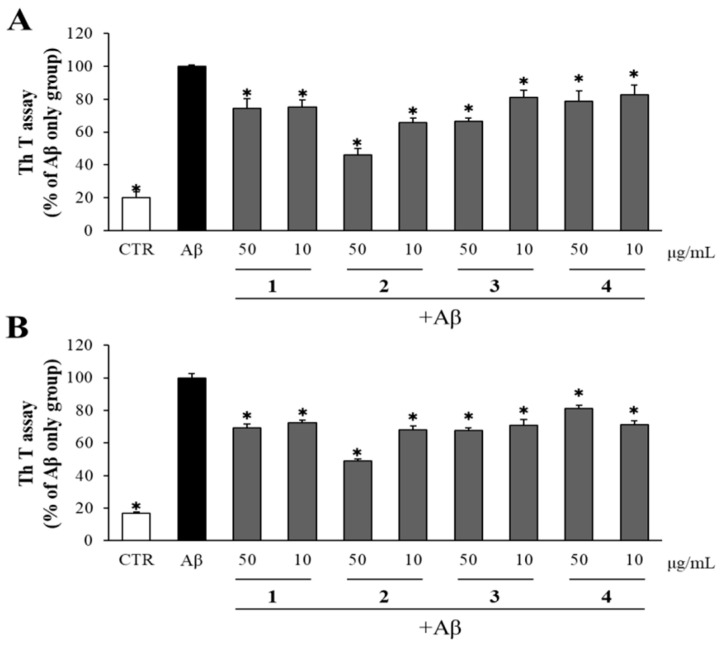
Inhibitory effect of isolated compounds on Aβ aggregation/disaggregation. (**A**) Aβ was incubated with 50 and 10 μg/mL of compounds isolated from DME. After 24 h, the Aβ aggregation was determined by Th T assay. (**B**) Aβ pre-aggregated for 24 h was incubated with 50 and 10 μg/mL of compounds isolated from DME. After 24 h, the Aβ disaggregation was determined by Th T assay. All data represent the means ± SD of three different experiments. * *p* < 0.05, significantly different from Aβ-only group.

**Figure 7 pharmaceuticals-16-01606-f007:**
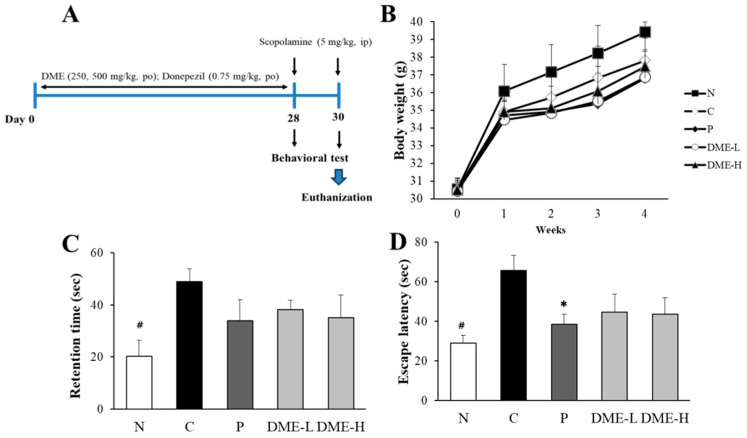
Effect of DME on scopolamine-induced memory impairment in mice. (**A**) A schematic illustration of the animal study. (**B**) The body weight of the mice was measured twice a week for 4 weeks. (**C**) The retention time staying in the dark room of the mice in the passive avoidance test was measured. (**D**) The escape latency, i.e., the time for the mice to find the escape platform in the Morris water maze test, was recorded up to 120 s. All data represent the mean ± SE (n = 9). * *p* < 0.05, # *p* < 0.005, significantly different from scopolamine-treated control groups (N; normal, C; saline + scopolamine, P; donepezil + scopolamine, DME-L; 200 mg/kg + scopolamine, DME-H; 500 mg/kg + scopolamine).

**Figure 8 pharmaceuticals-16-01606-f008:**
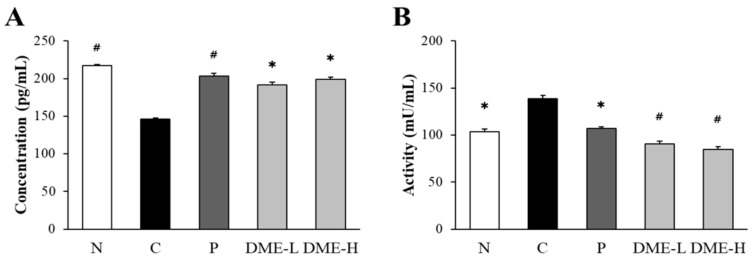
Effect of DME on the levels of ACh and AChE in mice. (**A**) The levels of ACh in the whole brains was determined by ELISA. (**B**) The activity of AChE in the whole brains of mice was determined by ELISA. All data represent the mean ± SE (n = 9). * *p* < 0.05, # *p* < 0.005, significantly different from scopolamine-treated control groups (N; normal, C; saline + scopolamine, P; donepezil + scopolamine, DME-L; 200 mg/kg + scopolamine, DME-H; 500 mg/kg + scopolamine).

**Figure 9 pharmaceuticals-16-01606-f009:**
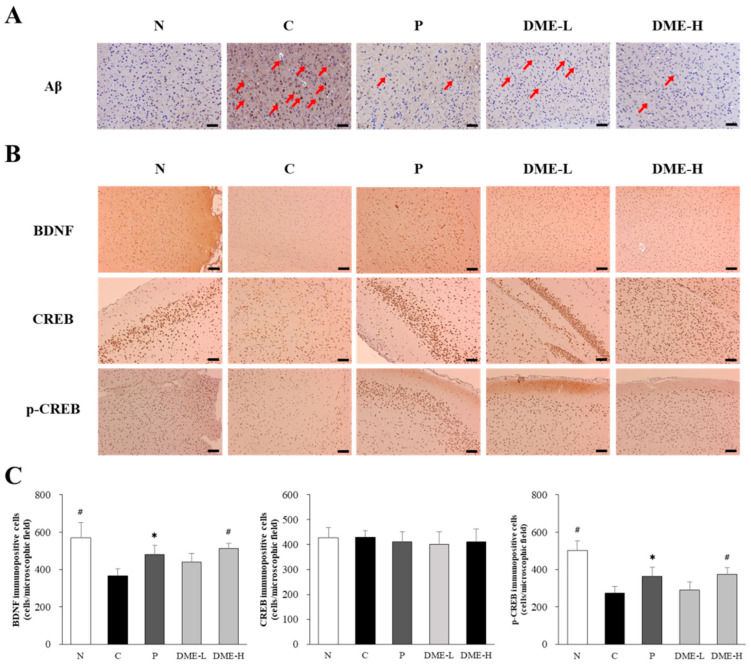
Effect of DME on Aβ deposition, BDNF, and CREB in the brains. Representative images of immunohistochemical analysis with anti-Aβ antibody (**A**), and anti-BDNF, anti-CREP, and anti-p-CREB antibodies (**B**) in the hippocampal region are shown (n = 7). Aβ depositions are indicated by arrows. (**C**) Immune-positive cells with BDNF, CREB, and p-CREB were quantified with Image J Software (1.5k version). Scale bar = 100 μm. All data represent the mean ± SE (n = 7). * *p* < 0.05, # *p* < 0.005, significantly different from scopolamine-treated control groups (N; normal, C; saline + scopolamine, P; donepezil + scopolamine, DME-L; 200 mg/kg + scopolamine, DME-H; 500 mg/kg + scopolamine).

## Data Availability

All data are contained within the article and Appendix A.

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
