# Peer review of "Neuroprotective Effects of *Davallia mariesii* Roots and Its Active Constituents on Scopolamine-Induced Memory Impairment in In Vivo and In Vitro Studies"

_pharmaceuticals, 2023, doi:10.3390/ph16111606_

Round 1

Reviewer 1 Report

Comments and Suggestions for Authors

Dear author,

Thanks for submitting your research manuscript entitled "Neuroprotective effects of Davallia mariesii roots and its active constituents on scopolamine-induced memory impairment in in vivo and in vitro studies".

From my experience in the field, this manuscript is dealing with very important aspect of Davallia mariesii roots and its active constituents on scopolamine-induced memory impairment.

I recommend the revision of this manuscript (with major comments) as it worth publishing and will add good information.

Before giving my final comments and revising this manuscript, the author must address the following comments scientifically.
Reviewer  concerns:

Please find out the following comments:

·         The rationale and purpose behind selecting Davallia mariesii roots and its active constituents on scopolamine-induced memory impairment is unclear and need to reframe in introduction and discussion.

·         Updates old & outdated references.

·         Major drawback is the lack of supporting references and incomplete experimental and paradigms.

·         A separate detail paragraph required here, to explain the experimental design with detail explanation in flow chart also.

·         How was the sample size determined? Ideally, a priori sample size calculation should be performed to determine the appropriate sample size.

·         Normality and variance homogeneity should be assessed across all groups of the same outcome variable and not individual experimental groups. If the data were not normally distributed or variance homogeneity was not met, nonparametric tests need to be performed. Parametric data should be reported as mean +/- sd, while nonparametric data should be given/displayed as median and interquartile range. Longitudinal data should be analyzed using repeated measures tests.

·         Provide the ethical approval number with date.

·         All results are very poorly explained. Revised explanation for all as above mentioned instructions.

-          Results need more clarification and significant justification. Differentiating between the outcome and the discussion sections is quite difficult.

-     To address the outcome of results separately, avoiding the disease condition and maintaining physiological condition. How they correlate with the existing literature, it would be better if the author restructured to take a more critical approach of Neuroprotective effects of Davallia mariesii roots and its active constituents on scopolamine-induced memory impairment in in vivo and in vitro studies.

- All histo images required scale and magnification on individual panel pictures. Also, provide the specific area with arrow/round color dots for further explanation.  

- Results need more clarification and significant justification. Differentiating between the outcome and the discussion sections is quite difficult.  

- High note: Must provide all results description and Use proper statistical reporting: i.e. for the results of each statistical test, the authors should report the statistical test that was applied, the test statistic (e.g. t, U, F, r), degrees of freedom as subscripts to the test statistic, and the exact probability value, including those for normality and variance homogeneity tests. Statistics should be reported in APA format, i.e.: t(df) = value, p = value; F(df1,df2) = value, p = value; r(df) = value, p = value; [chi]2 (df, N = value) = value, p = value; Z = value, p = value.  Include statements on the tests for normality and variance heterogeneity and respective results. If the data were not normally distributed or variance heterogeneity was not met, nonparametric tests need to be applied.

-     In the discussion and the conclusion, the aims, rationale, and future perspectives are not evident clearly in relation to in-vitro experimentation.
-     The discussion is usually unorganized at the beginning to address and evaluate all the observations at the end. It makes the results easier to contextualize and more straightforward to comprehend.

- Furthermore, a minimal critical analysis should be provided, along with current study limitations and the future perspective as separate paragraphs.

-          Need to revise the conclusion scientifically. Not accepted in its current form.

-          A detailed revision shortening, ordering and following the commented ideas could improve this paper.

-          Several typewriting mistakes are present and need correction. This reviewer remains at the entire disposal for the next version.

Comments on the Quality of English Language

Extensive editing of English language required

Author Response

Reviewer 1
Reviewer  concerns:
Please find out the following comments:

1. The rationale and purpose behind selecting Davallia mariesii roots and its active constituents on scopolamine-induced memory impairment is unclear and need to reframe in introduction and discussion.

--- As suggested by the reviewer, the rationale for selecting D. mariesii rooots and the scopolamine-injected mouse model has been included in the Introduction section (page 2, 64-80).

2. Updates old & outdated references.
· Major drawback is the lack of supporting references and incomplete experimental and paradigms.
--- The references have been reviewed and updated throughout the manuscript.

· A separate detail paragraph required here, to explain the experimental design with detail explanation in flow chart also.

--- A schematic illustration of the animal study has been added in Figure 7A, along with a detailed explanation for the experiment in the Materials and Methods section (page 14, 501-511).

· How was the sample size determined? Ideally, a priori sample size calculation should be performed to determine the appropriate sample size.

--- The sample size consisted of 9 mice per group. While it is ideal to conduct a preliminary study, the institution that conducted the animal study recommanded 9 mice per group based on their experience.

· Normality and variance homogeneity should be assessed across all groups of the same outcome variable and not individual experimental groups. If the data were not normally distributed or variance homogeneity was not met, nonparametric tests need to be performed. Parametric data should be reported as mean +/- sd, while nonparametric data should be given/displayed as median and interquartile range. Longitudinal data should be analyzed using repeated measures tests.

--- In this study, the test samples (treated groups) consist of independent simple random samples. The statistical analysis involved several steps. Initially, we conducted a variance test using Levene's test. If the p-value exceeded 0.05, we accepted the assumption of homoscedasticity. If the p-value was less than 0.05, we rejected the assumption of homoscedasticity (indicating heteroscedasticity). Subsequently, based on these results, when homoscedasticity was assumed, we proceeded with the evaluation using one-way ANOVA with the Fisher's Least Significant Difference test tests. Conversely,in casess where the assumption of homoscedasticity was not met (indicating heteroscedasticity), we analyzed the data using the Kruskal-Wallis test and Mann-Whitney U-test. It is important to note that all the statistical analyses were conducted in accordance with the reviewer's comment, and this information has been incorporated intro the Statistical analysis section (page 16, 594-601).

· Provide the ethical approval number with date.

--- The date has been included in the section 4.6 (page 14, 510 and page 16, 614).

· All results are very poorly explained. Revised explanation for all as above mentioned instructions.
Results need more clarification and significant justification. Differentiating between the outcome and the discussion sections is quite difficult.

--- In response to the reviewer's suggestions, the results have been revised and explained in greater detail (highlighted in red).

-To address the outcome of results separately, avoiding the disease condition and maintaining physiological condition. How they correlate with the existing literature, it would be better if the author restructured to take a more critical approach of Neuroprotective effects of Davallia mariesii roots and its active constituents on scopolamine-induced memory impairment in in vivo and in vitro studies.

--- In response to the reviewer's comments, the Discusion section has been meticulously revised (highlighted in red).

- All histo images required scale and magnification on individual panel pictures. Also, provide the specific area with arrow/round color dots for further explanation. 

--- As per the reviewer's suggestion, scale bars havev been added to Figure 9A & B, and quantification results are now included in Figure 9C. The results are explained in the section 2.9 (page 3, 303-317).

- Results need more clarification and significant justification. Differentiating between the outcome and the discussion sections is quite difficult.  

--- The results have been further refined and elaborated upon (highlighted in red).

- High note: Must provide all results description and Use proper statistical reporting: i.e. for the results of each statistical test, the authors should report the statistical test that was applied, the test statistic (e.g. t, U, F, r), degrees of freedom as subscripts to the test statistic, and the exact probability value, including those for normality and variance homogeneity tests. Statistics should be reported in APA format, i.e.: t(df) = value, p = value; F(df1,df2) = value, p = value; r(df) = value, p = value; [chi]2 (df, N = value) = value, p = value; Z = value, p = value.  Include statements on the tests for normality and variance heterogeneity and respective results. If the data were not normally distributed or variance heterogeneity was not met, nonparametric tests need to be applied.

--- In this study, the test samples (treated groups) consist of independent simple random samples. The statistical analysis involved several steps. Initially, we conducted a variance test using Levene's test. If the p-value exceeded 0.05, we accepted the assumption of homoscedasticity. If the p-value was less than 0.05, we rejected the assumption of homoscedasticity (indicating heteroscedasticity). Subsequently, based on these results, when homoscedasticity was assumed, we proceeded with the evaluation using one-way ANOVA with the Fisher's Least Significant Difference test tests. Conversely,in casess where the assumption of homoscedasticity was not met (indicating heteroscedasticity), we analyzed the data using the Kruskal-Wallis test and Mann-Whitney U-test. It is important to note that all the statistical analyses were conducted in accordance with the reviewer's comment, and this information has been incorporated intro the Statistical analysis section (page 16, 594-601).

-In the discussion and the conclusion, the aims, rationale, and future perspectives are not evident clearly in relation to in-vitro experimentation.

--- In accordance with the reviewer's comments, the aims, rationale, and future perspectives have been carefully revised in the Conclusion section.

- The discussion is usually unorganized at the beginning to address and evaluate all the observations at the end. It makes the results easier to contextualize and more straightforward to comprehend.

--- As suggested by the reviewer, the Discusion section has also been throughly revised (highlighted in red).

- Furthermore, a minimal critical analysis should be provided, along with current study limitations and the future perspective as separate paragraphs.
- Need to revise the conclusion scientifically. Not accepted in its current form.
- A detailed revision shortening, ordering and following the commented ideas could improve this paper.
- Several typewriting mistakes are present and need correction. This reviewer remains at the entire disposal for the next version.

--- In accordance with the reviewer's comments, the manuscript has been thoughtfully revised to address the study's limitations, which are explained in the Discussio section (page 12, 421-424).

Reviewer 2 Report

Comments and Suggestions for Authors

The manuscript entitled "Neuroprotective effects of Davallia mariesii roots and its active constituents on scopolamine-induced memory impairment in in vivo and in vitro studies”. However, a few comments and suggestions must be addressed before this reviewer recommends the publication of this work in the Journal.

Comments

1.       What is the significance of beta-amyloid (Aβ) proteins in the context of Alzheimer's disease (AD), and how do they contribute to the disease?

2.       The author should explain, how did the ethanol extract of D. mariesii roots and its active constituents impact the production and aggregation of Aβ, as described in the text?

3.       Can you explain the changes in the levels of acetylcholine, acetylcholinesterase, BDNF, and p-CREB observed in the brains of mice treated with the extract of D. mariesii roots?

4.       What are some possible implications or applications of this research in the development of therapeutic or preventative agents for AD?

5.       What evidence is provided to support the claim that all isolated compounds reduced the expression of Aβ and β-secretase in a dose-dependent manner?

6.       Based on the results presented, what are the potential therapeutic and preventative applications of DME and its active constituents for AD?

7.       The author should revise the outstanding points and highlights of this work in the conclusion.

8.       Typographical errors and superfluous spaces throughout the manuscript should be corrected.

Author Response

Reviewer 2
The manuscript entitled "Neuroprotective effects of Davallia mariesii roots and its active constituents on scopolamine-induced memory impairment in in vivo and in vitro studies”. However, a few comments and suggestions must be addressed before this reviewer recommends the publication of this work in the Journal.

Comments

1.       What is the significance of beta-amyloid (Aβ) proteins in the context of Alzheimer's disease (AD), and how do they contribute to the disease?

--- In response to the reviewer's suggestioion, the significance of beta-amyloid in AD has been incorporated into both the Introduction and Discussion sections (page 2, 53-64 & page 11, 325-337).

2.       The author should explain, how did the ethanol extract of D. mariesii roots and its active constituents impact the production and aggregation of Aβ, as described in the text?

        --- The potential mechanisms of DME and its compounds have been elucidated in the Duscission section (page 11, 342-354).

3.       Can you explain the changes in the levels of acetylcholine, acetylcholinesterase, BDNF, and p-CREB observed in the brains of mice treated with the extract of D. mariesii roots?

--- In response to the reviewer's suggestioion, the explanations the changes in ACh, AChE, BDNF and p-CREB provided in the sections 2.8 (page 9, 275-293) and 2.9 (page 10, 303-317). 

4.       What are some possible implications or applications of this research in the development of therapeutic or preventative agents for AD?

--- The potential applications of DME has been addressed in the Conclusion section (page .
16, 600-604)

5.       What evidence is provided to support the claim that all isolated compounds reduced the expression of Aβ and β-secretase in a dose-dependent manner?

--- As per the reviewer's precise observation, the effects of compound 4 was found not to be dose-dependent. Consequently, the phrase “in a dose-dependent manner” has been removed from section 2.5.

6.       Based on the results presented, what are the potential therapeutic and preventative applications of DME and its active constituents for AD?

--- The potential applications of DME have been addressed in the Conclusion section (page .
16, 600-604).

7.       The author should revise the outstanding points and highlights of this work in the conclusion.

--- We appreciate the reviewer's valuable suggestion, and the Conclusion section has been revised accordingly (page 16, 600-604).

8.       Typographical errors and superfluous spaces throughout the manuscript should be corrected.

--- As noted by the reviewer, the manuscript has been throughly reviewed by a native English speaker.

Reviewer 3 Report

Comments and Suggestions for Authors

The manuscript by Chung Hyeon Lee et et., entitled "Neuroprotective effects of Davallia mariesii roots and its active constituents on scopolamine-induced memory impairment in in vivo and in vitro studies" a research article reported that the Davallia mariesii roots in in-vivo and in vitro AD model. Overall, the paper is interesting and well-written. The authors have identified an interesting scientific question, but significant issues are present that must be addressed.

Comments:

1.       The authors should include the schematic diagram for the in-vivo experimental design.

2.       The authors should give the detailed procedure for the MWM test. How many days the mice were trained for the test?

3.       In Immunohistochemistry images the authors mentioned 100x magnification. But, the images do not look 100x magnification. The authors should ensure that magnification and should include the scale bar in the image.

4.       In Figure 9, the authors reported that DME treatment reduces the A-beta and increases the BDNF, CREB, and p-CREB. How do the authors analyze the intensity? The authors should represent the intensity as a graph for each marker expression level.

5.       The authors should include the molecular weight (MW) for each Western blot image.

6.       The authors should include the detailed procedure for the Western blot image quantification.

Author Response

Reviewer 3

The manuscript by Chung Hyeon Lee et et., entitled "Neuroprotective effects of Davallia mariesii roots and its active constituents on scopolamine-induced memory impairment in in vivo and in vitro studies" a research article reported that the Davallia mariesii roots in in-vivo and in vitro AD model. Overall, the paper is interesting and well-written. The authors have identified an interesting scientific question, but significant issues are present that must be addressed.
Comments:
1.        The authors should include the schematic diagram for the in-vivo experimental design.
--- In response to the reviewer's comments, a schematic diagram for the in vivo study has been included in Figure 7A.

2.        The authors should give the detailed procedure for the MWM test. How many days the mice were trained for the test?
--- As suggested by the reviewer, detailed procedures for the MWM test and training days (1 day) have been added to the Material and Method section (page 15, 530-542).

3.        In Immunohistochemistry images the authors mentioned 100x magnification. But, the images do not look 100x magnification. The authors should ensure that magnification and should include the scale bar in the image.
--- The magnification of images has been rechecked, and scale bars have been added to Figure 9A & B, along with explanation in the figure legend

4.        In Figure 9, the authors reported that DME treatment reduces the A-beta and increases the BDNF, CREB, and p-CREB. How do the authors analyze the intensity? The authors should represent the intensity as a graph for each marker expression level.
--- In accordance with the reviewer's feedback, graphs displaying the intensity of marker expression have been introduced in Figure 9C and are explained in the Result section (page 10, 310-317).

5.        The authors should include the molecular weight (MW) for each Western blot image.
--- As the reviewer pointed out, melecular weights have been incorporated into the western blot images in Figures 1, 2, and 5.

6.        The authors should include the detailed procedure for the Western blot image quantification.
--- The procedure for the quantification of western blot images is now provided in the Material and Methods section (page 14, 479-482).

Reviewer 4 Report

Comments and Suggestions for Authors

The authors demonstrated that the extracts of D. mariesii roots have neuroprotective effects on AD and recover scopolamine-induced memory impairment in vitro and in vivo. However, there are a few flaws.

1. In figure 1, figure 2 and figure 5, the extracts of D. mariesii roots can effectively decrease the production of β-secretase and βAPP. But, it is not clear that the decrease of β-secretase and βAPP is caused by the inhibition of the activity of β-secretase or by the production APP. A western blot of APP or qPCR of APP can elucidate this issue.

2. It is quite nice that some of the constitutions of the D. mariesii roots can also inhibit the aggregation of Aβ. However, only steady-state Th-T assay cannot fully convince the inhibitory effect. An extra study such as TEM images are necessary.  

3. The use of scopolamine to induce Alzheimer’s disease should be introdiced in discussion at least. 

Comments on the Quality of English Language

A few typos need to be revised. 

Author Response

Reviewer 4

The authors demonstrated that the extracts of D. mariesii roots have neuroprotective effects on AD and recover scopolamine-induced memory impairment in vitro and in vivo. However, there are a few flaws.
 1. In figure 1, figure 2 and figure 5, the extracts of D. mariesii roots can effectively decrease the production of β-secretase and βAPP. But, it is not clear that the decrease of β-secretase and βAPP is caused by the inhibition of the activity of β-secretase or by the production APP. A western blot of APP or qPCR of APP can elucidate this issue.
--- We appreciate the reviewer's comments. This aspect had not been considered. 
If DME downregulates APP expression, it would lead to a decrease in sAPPβ production. However, unless DME completely blocks the expression of APP, the produced APP can still be cleaved into sAPPβ by β-secretase. In this case, if DME reduces the levels of β-secretase, it would result in decreased sAPPβ production. Therefore, the reduction in sAPPβ production may be affected and influenced by the decreased expression of APP, but it could be just one part of the mechanism. Therefore, at least a portion of the mechnism behind the reduced sAPPβ production is likely through the inhibition of β-secretase. Nonetheless, it is noteworthy to consider measuring the expression of APP. This limitation has been added in the Discussion section (page 12, 422-425) 
 2. It is quite nice that some of the constitutions of the D. mariesii roots can also inhibit the aggregation of Aβ. However, only steady-state Th-T assay cannot fully convince the inhibitory effect. An extra study such as TEM images are necessary.  
--- Th T is a commonly used probe for monitoring in vitro amyloid fibril formation, and DME not only inhibited Aβ production, but also Aβ aggregation. While it would be preferable to provide TEM images, our institution lacks the facility, and we lack experience in microscopic studies. As suggested by the reviewer, these data could be useful for further developing DME as a functional food (which we are planning to do). Therefore, we will include TEM imaging in a future study through collaboration with an expert group. The limitation of this study has been added to the Discussion section (page 11, 350-354)

3. The use of scopolamine to induce Alzheimer’s disease should be introdiced in discussion at least. 
 --- We appreciate the reviewer's suggestion. A detailed explanation of the scopolamine-administered animal study has been included in both the Introduction and discussion section (page 2, 64-73 & page 12, 385-388).

Reviewer 5 Report

Comments and Suggestions for Authors

Major comments

In this study, the authors tried to elucidate novel beneficial effects of ethanol extract of Davallia mariesii roots (DME) on Alzheimer’s disease (AD)-related pathologies. The findings are potentially interesting and important; however, there are issues of serious concerns.

1.         Figure 9

The data provided are immunohistochemical images only. Since there are no quantitative and statistical analyses regarding amyloid-β, BDNF, CREB, and p-CREB between the groups, it remains unclear whether DME has statistically significant beneficial effects on these factors. In this respect, current states of Figure 9 do not support the notion described in the Discussion and Conclusions sections.

2.         The Discussion section

The authors just summarized findings of previous studies concerning bioactivities of flavonoids, acetylcholine, and BDNF. I suggest that the authors provide deeper discussions about novelty and potential mechanisms underlying effects of DME.

3.         In vivo model

There is no explanation about characteristics of scopolamine-administered mice. I recommend that the authors provide details in the Introduction section with the relevant citations. For instance, scopolamine increases the expression levels of amyloid-β in the brain and scopolamine-administered mice are used as one of the mouse models of AD.

4.         The Conclusions section

The description “The memory impairment induced by scopolamine was attenuated by DME (line 514)” is not appropriate, because DME did not exert statistically significant beneficial effects on cognitive impairment as shown in Figure 7 B and C. The authors should revise the sentence.

5.         The Supplementary data

The authors explain the data on the supplementary Figures (e.g., Figure S1 [line 86]), but the supplementary information is not available for the reviewer. My concern is that those figures were not uploaded appropriately at the submission; if so, such a situation should be avoided.

Minor comments

1.         The sentence “In addition, the… (lines 55–57)” needs the relevant citations.

2.         Abbreviations (Hx, DCM, EA, and DW [lines 104–105] and CCs [line 151]) must be spelled out completely on initial appearance in text. Similarly, CTR, PC, and DME in the Figure 1 should be spelled out completely in the legends. I suggest that the authors check the text and all the figures.

3.         I suggest that the title of Figure 4 is described accurately as follows: The chemical structures of the isolated compounds from ethyl acetate fraction of DME.

4.         The tissues analyzed on AChE and ACh should be specified in the text of the subsection 2.8., legends of Figure 8, and the subsection 4.9., although the authors described “in the brains of mice (line 25)” in the Abstract section.

5.         The number of experiments and/or mice should be described in the legends of Figure 9.

6.         I suggest that the authors provide discussion on the permeability of the blood–brain barrier to compounds of DME.

7.         The authors should provide details on the genetic manipulation for the stable expression of amyloid precursor protein in Chinese Hamster Ovary cells in the Materials and Methods section. Alternatively, the relevant citations should be provided.

8.         I recommend that the authors check whether “1 μg/g (line 372)” is correct.

9.         I suggest that the amount of protein used in Western blot is provided in the Materials and Methods section.

10.     Administration route of DME for in vivo experiments should be specified in the Materials and Methods section.

11.     I suggest that the authors provide more details on the process to obtain test samples for ELISA, such as homogenization of the brain, in the subsection 4.9.

12.     The incubation period of antibodies should be described in the subsection 4.10.

Author Response

Reviewer 5

Major comments
In this study, the authors tried to elucidate novel beneficial effects of ethanol extract of Davallia mariesii roots (DME) on Alzheimer’s disease (AD)-related pathologies. The findings are potentially interesting and important; however, there are issues of serious concerns.
 1.         Figure 9
The data provided are immunohistochemical images only. Since there are no quantitative and statistical analyses regarding amyloid-β, BDNF, CREB, and p-CREB between the groups, it remains unclear whether DME has statistically significant beneficial effects on these factors. In this respect, current states of Figure 9 do not support the notion described in he Discussion and Conclusions sections.
--- As the reviewer commented, graphs showing the intensity of marker expression have been added to Figure 9C and explained in the Result section (page 10, 310-317).
2.         The Discussion section
The authors just summarized findings of previous studies concerning bioactivities of flavonoids, acetylcholine, and BDNF. I suggest that the authors provide deeper discussions about novelty and potential mechanisms underlying effects of DME.
--- We appreciate the reviewer's suggestion. The Discussion section has been throughly revised based on the reviewer's comments (highlightened red).
3.         In vivo model
There is no explanation about characteristics of scopolamine-administered mice. I recommend that the authors provide details in the Introduction section with the relevant citations. For instance, scopolamine increases the expression levels of amyloid-β in the brain and scopolamine-administered mice are used as one of the mouse models of AD.
 --- We appreciate the reviewer's suggestion. A detailed explanation of the scopolamine-administered animal study has been added to the Introduction section (page 2, 64-73).
4.         The Conclusions section
The description “The memory impairment induced by scopolamine was attenuated by DME (line 514)” is not appropriate, because DME did not exert statistically significant beneficial effects on cognitive impairment as shown in Figure 7 B and C. The authors should revise the sentence.
--- As the reviewer accurately pointed out, the sentense has been carefully revised (page 16, 592-605).

5.         The Supplementary data
The authors explain the data on the supplementary Figures (e.g., Figure S1 [line 86]), but the supplementary information is not available for the reviewer. My concern is that those figures were not uploaded appropriately at the submission; if so, such a situation should be avoided.
 --- I appologize for the mistake. We ahve checked the correct uploading of the figures.

Minor comments
1.         The sentence “In addition, the… (lines 55–57)” needs the relevant citations.
     --- The reference (Varadarajan et al., 2000) has been added to the text. 

2.         Abbreviations (Hx, DCM, EA, and DW [lines 104–105] and CCs [line 151]) must be spelled out completely on initial appearance in text. Similarly, CTR, PC, and DME in the Figure 1 should be spelled out completely in the legends. I suggest that the authors check the text and all the figures.
--- As the reviewer suggested, the abbrevations have been spelled out upon their initial appearance in the text (DME, Hx, DCM, EA, DE, CTR, PC, CC).

3.         I suggest that the title of Figure 4 is described accurately as follows: The chemical structures of the isolated compounds from ethyl acetate fraction of DME.
--- The title of Figure 4 has been accurately described. 

4.         The tissues analyzed on AChE and ACh should be specified in the text of the subsection 2.8., legends of Figure 8, and the subsection 4.9., although the authors described “in the brains of mice (line 25)” in the Abstract section.
    --- Regarding the measurement of AChE and ACh, we used whole brains in this experiment due to the difficulty in dissecting the hippocampus. This has been specified in subsection 2.8 (page 9, 276), figure 8, subsection 4.9 (page 15, 550-551) and abstract (page 1, 28).

5.         The number of experiments and/or mice should be described in the legends of Figure 9.
    --- The number of experiments (n = 7) has been added to the legend of Figure 9.

6.         I suggest that the authors provide discussion on the permeability of the blood–brain barrier to compounds of DME.
    --- Thank you for the suggestion. The permeanility issue has been added to the Discussion section (page 12, 374-381).

7.         The authors should provide details on the genetic manipulation for the stable expression of amyloid precursor protein in Chinese Hamster Ovary cells in the Materials and Methods section. Alternatively, the relevant citations should be provided.
    --- The reference for APP-CHO cells stably expressing APP has been added (page 13, 446).

8.         I recommend that the authors check whether “1 μg/g (line 372)” is correct.
    --- I apologize for the error. It has been corrected to 50 μg/mL.

9.         I suggest that the amount of protein used in Western blot is provided in the Materials and Methods section.
--- The amount of proteins loaded (20 ug) has been added to the text. 

10.     Administration route of DME for in vivo experiments should be specified in the Materials and Methods section.
    --- Thank you for pointing this out. The adminstration route has been correctly added in the Materials and Methods section (page 13, 506-516). 

11.     I suggest that the authors provide more details on the process to obtain test samples for ELISA, such as homogenization of the brain, in the subsection 4.9.
    --- As the reviewer suggested, the detailed homogenization procedure has been added in the subsection 4.9 (page 13, 546-552).

12.     The incubation period of antibodies should be described in the subsection 4.10.
    --- As the reviewer suggested, the incubation temperature and time have been added in the subsection 4.9 (page 13, 573-574).

Round 2

Reviewer 1 Report

Comments and Suggestions for Authors

Dear author,

After careful revision, manuscript revised successfully, and can be proceed further for publication.

Comments on the Quality of English Language

Minor editing of English language required

Reviewer 4 Report

Comments and Suggestions for Authors

The revised manuscript has excellent improvement. 

Reviewer 5 Report

Comments and Suggestions for Authors

I think that the authors appropriately responded to comments raised in the 1st round of peer-review. I suggest that the authors provide statistical significance in Supplementary Figure S2.